# AnyAttack: Towards Large-scale Self-supervised Generation of Targeted Adversarial Examples for Vision-Language Models

## Abstract

Due to their multimodal capabilities, Vision-Language Models (VLMs) have found numerous impactful applications in real-world scenarios. However, recent studies have revealed that VLMs are vulnerable to image-based adversarial attacks, particularly targeted adversarial images that manipulate the model to generate harmful content specified by the adversary. Current attack methods rely on predefined target labels to create targeted adversarial attacks, which limits their scalability and applicability for large-scale robustness evaluations. In this paper, we propose **AnyAttack**, a self-supervised framework that generates targeted adversarial images for VLMs without label supervision, allowing **any** image to serve as a target for the **attack**. To address the limitation of existing methods that require label supervision, we introduce a contrastive loss that trains a generator on a large-scale unlabeled image dataset, LAION-400M dataset, for generating targeted adversarial noise. This large-scale pre-training endows our method with powerful transferability across a wide range of VLMs. Extensive experiments on five mainstream open-source VLMs (CLIP, BLIP, BLIP2, InstructBLIP, and MiniGPT-4) across three multimodal tasks (image-text retrieval, multimodal classification, and image captioning) demonstrate the effectiveness of our attack. Additionally, we successfully transfer AnyAttack to multiple commercial VLMs, including Google's Gemini, Claude's Sonnet, and Microsoft's Copilot. These results reveal an unprecedented risk to VLMs, highlighting the need for effective countermeasures. Upon publication, we will release the pre-trained generator to support further research in addressing this challenge.

## 1 Introduction

Vision-Language Models (VLMs) have exhibited remarkable performance across a diverse array of tasks, primarily attributed to the scale of training data and model size (Radford et al., 2021; Li et al., 2023; Zhu et al., 2024). Despite their remarkable performance, these models, heavily reliant on visual inputs, remain vulnerable to image-based adversarial attacks[1], which are carefully crafted input images designed to mislead the model into making incorrect predictions (Szegedy et al., 2013). While general, untargeted adversarial attacks only aim to induce incorrect outputs, *targeted adversarial attacks* present a more insidious threat, manipulating the model's output to yield an *adversary-specified, predetermined response*. For instance, a benign image such as a landscape could be subtly altered to elicit harmful text descriptions such as "violence" or "explicit content" from the model. Such manipulation could have severe implications for content moderation systems, potentially leading to the removal of legitimate content or the inappropriate distribution of harmful material.

Both targeted adversarial attacks and jailbreak techniques (Zou et al., 2023; Bagdasaryan et al., 2023; Carlini et al., 2024) aim to induce harmful responses from models. However, the key distinction lies in their outputs: targeted adversarial attacks produce adversary-specified, *predetermined* responses, whereas jailbreak attacks elicit *non-predetermined* responses. We will further elaborate on this dis-

---

[1]For simplicity, we will refer to image-based adversarial attacks as "adversarial attacks" in the remainder of this paper, distinguishing them from text-based adversarial attacks.

Figure 1: (a) Existing methods generate targeted adversarial attacks guided by the target labels. (b) Our AnyAttack is a self-supervised attack method that does not require any labels. The dashed lines highlight the key differences between the two approaches.

tinction in Section 2. As VLMs become increasingly accessible to the public, facilitating the rapid proliferation of downstream applications, this vulnerability poses a significant threat to the reliability and security of VLMs in real-world scenarios. Therefore, exploring new targeted attack methods tailored to VLMs is crucial to address these vulnerabilities. However, existing targeted attack methods on VLMs present challenges due to the reliance on target labels for supervision, which limits the scalability of the training process. For example, it is impractical to expect a generator trained on ImageNet (Russakovsky et al., 2015) to produce effective adversarial noise for VLMs.

To address this limitation, we propose a novel self-supervised learning framework called **AnyAttack**, which utilizes the original image itself as supervision, allowing **any** image to serve as a target for the deployment of targeted adversarial **attacks**. Figure 1 illustrates the differences between our framework and the existing methods. Drawing inspiration from popular contrastive learning techniques (Chen et al., 2020; He et al., 2020; Radford et al., 2021; Li et al., 2023), we introduce a contrastive loss to allow training a generator on the large-scale dataset to generate targeted adversarial noise. This encourages the generated noise to emulate the original image itself (positive pairs) in the embedding space while differentiating from other images (negative pairs). The unsupervised nature of our method enables large-scale pre-training of our generator on the LAION-400M dataset (Schuhmann et al., 2021), allowing for the generation of effective adversarial noise. Benefiting from this large-scale pre-training and exposure to a broader spectrum of images, our method shows powerful transferability across a wide range of VLMs. To the best of our knowledge, this is the first time such a large-scale dataset has been employed to train a generator for the generation of targeted adversarial noise. Notably, the generator can produce adversarial images with any given image, imposing no restrictions on the image itself, thereby introducing an unprecedented security risk to the entire community.

Furthermore, to enable adaptation to specific domains and multimodal tasks, we fine-tune the pre-trained generator on downstream datasets for adapting downstream vision-language tasks. To demonstrate the effectiveness of our approach, we conduct extensive experiments on 5 target VLMs (CLIP (Radford et al., 2021), BLIP (Li et al., 2022), BLIP2 (Li et al., 2023), InstructBLIP (Dai et al., 2023), and MiniGPT-4 (Zhu et al., 2024)), across 3 multimodal tasks (image-text retrieval, multimodal classification, and image captioning). We also evaluate our method on commercial VLMs, including Google's Gemini, Claude's Sonnet, and Microsoft's Copilot.

In summary, our main contributions are:

- We propose **AnyAttack**, a novel self-supervised framework for generating targeted adversarial attacks on VLMs without the need for target labels, allowing **any** image to be used as a target for the **attack**.

- We introduce a contrastive loss that facilitates the training of the generator for adversarial noise using unlabeled, large-scale datasets. This makes our **AnyAttack** framework scalable, effectively overcoming the limitations of previous methods.

- We demonstrate the effectiveness of our AnyAttack on five mainstream open-source VLMs across three multimodal tasks. Additionally, we successfully transfer our attack to three commercial VLMs. These results offer valuable insights into the vulnerabilities of state-of-the-art models in real-world applications.

## 2 RELATED WORK

**Targeted Adversarial Attacks**   A number of works have been proposed to enhance the effectiveness and transferability of targeted adversarial attacks against vision models. Input augmentation techniques like image translation (Dong et al., 2019), cropping (Wei et al., 2023), mixup (Wang et al., 2021; Liu & Lyu, 2024), and resizing (Xie et al., 2019), have been employed to increase the diversity of adversarial inputs, thereby improving their transferability across different target models. Additionally, adversarial fine-tuning and model enhancement techniques have been explored to bolster the attack capabilities of surrogate models (Springer et al., 2021; Zhang et al., 2023; Wu et al., 2024). These methods typically involve retraining the surrogate models with a mix of clean and adversarial examples to improve their robustness against future attacks. Furthermore, optimization techniques have evolved to stabilize the update processes during adversarial training. Methods such as adaptive learning rates and gradient clipping have been integrated to ensure more consistent updates and enhance the overall performance of the adversarial attacks (Dong et al., 2018; Wang & He, 2021; Lin et al., 2023). These advancements collectively contribute to the development of more effective and transferable adversarial attacks in the realm of vision models.

**Jailbreak Attacks on VLMs**   VLMs have revolutionized DNNs by leveraging large-scale pre-training on diverse image-text datasets. These models learn to integrate visual and textual information effectively, enabling superb performance across a wide range of tasks. Broadly, VLMs can be categorized into two types: the first offers multimodal functionalities built on large language models (LLMs), complemented by visual models, such as BLIP2 (Li et al., 2023), InstructBLIP (Dai et al., 2023), and MiniGPT-4 (Zhu et al., 2024). The second type provides a more balanced approach, bridging textual and visual modalities efficiently, as seen in models like CLIP (Radford et al., 2021), ALIGN (Jia et al., 2021), and BLIP (Li et al., 2022). Recent advancements in VLMs have spurred parallel research into their vulnerabilities, with jailbreaks and adversarial attacks emerging as distinct threat vectors. Multimodal jailbreaks primarily exploit cross-modal interaction vulnerabilities in VLMs, with the intention of influencing LLMs (Zou et al., 2023). These attacks manipulate inputs of text (Wu et al., 2023), images (Carlini et al., 2024; Gong et al., 2023; Qi et al., 2024; Niu et al., 2024), or both simultaneously (Ying et al., 2024; Wang et al., 2024), aiming to elicit harmful but *non-predefined* responses. In contrast, image-based adversarial attacks focus on manipulating the image encoder of VLMs, typically leaving the LLM component largely undisturbed. The objective is to induce adversary-specified, *predetermined* responses through precise visual manipulations. Understanding these differences is crucial for explaining our methodology.

**Adversarial Attacks on VLMs**   Adversarial research on VLMs is relatively limited compared to the extensive studies on vision models, with the majority of existing attacks focusing primarily on untargeted attacks. Co-Attack (Zhang et al., 2022) was among the first to perform white-box untargeted attacks on several VLMs. Following this, more approaches have been proposed to enhance adversarial transferability for black-box untargeted attacks (Lu et al., 2023; Zhou et al., 2023; Yin et al., 2024; Xu et al., 2024). Cross-Prompt Attack (Luo et al., 2024) investigates a novel setup for adversarial transferability based on the prompts of language models. AttackVLM (Zhao et al., 2024) is the most closely related work, using a combination of text inputs and popular text-to-image models to generate guided images for creating targeted adversarial images. Although their approach shares a similar objective with our work, our method distinguishes itself by being self-supervised and independent of any text-based guidance.

## 3 PROPOSED ATTACK

In this section, we first present the preliminaries on CLIP and targeted adversarial attacks and then introduce our proposed AnyAttack and its two phases (i.e., pre-training and fine-tuning).

### 3.1 PROBLEM FORMULATION

**Threat Model**   This work focuses on transfer-based black-box attacks, where the adversary generates an adversarial image $x'$ using a fully accessible pre-trained surrogate model $f_s$. The adversary has no knowledge of the target VLMs $f_t$, including its architecture and parameters, nor can they

Table 1: The supervision and formulation of different targeted adversarial attack strategies. The first two rely on explicit target supervision (label or image), whereas our AnyAttack is unsupervised.

| Strategy | Formulation | |
|---|---|---|
| Target label supervision | $\min \mathcal{L}(f_s(x + \delta), f_s(y_t))$, | s.t. $y_t \neq y$ |
| Target image supervision | $\min \mathcal{L}(f_s(x + \delta), f_s(x_t))$, | s.t. $y_t \neq y$ |
| **AnyAttack (ours)** | $\min \mathcal{L}(f_s(\delta + x_r), f_s(x))$, | s.t. $x_r \neq x$ |

leverage the outputs of $f_t$ to reconstruct adversarial images. The adversary's objective is to cause the target VLM $f_t$ to incorrectly match the adversarial image $x'$ with the target text description $y_t$.

We begin by formulating the problem of targeted adversarial attacks. Let $f_s$ represent a pre-trained surrogate model, and $\mathcal{D} = \{(x, y)\}$ denote the image dataset, where $x$ is the original image and $y$ is the corresponding label (description). The attacker's objective is to craft an adversarial example $x' = x + \delta$ that misleads the target model $f_t$ into predicting a predefined target label $y_t$. In the context of VLMs, this objective requires that $x'$ aligns with $y_t$ as a valid image-text pair. The process of generating targeted adversarial images typically involves finding a perturbation $\delta$ using the surrogate model $f_s$. Existing strategies can be approached through two primary strategies: target label supervision and target image supervision.

The first approach utilizes the target label $y_t$ as supervision, directing the embedding of the adversarial image $x'$ to align with that of $y_t$, as demonstrated in AttackVLM-it Zhao et al. (2024). The second approach employs the target image $x_t$, which corresponds to $y_t$, as supervision to encourage the embedding of $x'$ to replicate that of $x_t$. This is illustrated in AttackVLM-ii (Zhao et al., 2024) and certain image-based attacks (Wei et al., 2023; Wu et al., 2024). Both methods depend on explicit target supervision, as summarized in Table 1. In these approaches, $\mathcal{L}$ denotes a distance-based loss function, such as Euclidean distance or cosine similarity. In contrast, our method, AnyAttack, employs the input image itself to guide the attack and thus is unsupervised. In this context, $x_r$ is a random image that is unrelated to $x$, while the adversarial noise $\delta$ is designed to align with the original image $x$ within the surrogate model's embedding space. In summary, existing methods generate adversarial noise that mimics other images, whereas our approach produces adversarial noise that closely resembles the original image itself.

## 3.2 ANYATTACK

**Framework Overview** Our proposed framework, **AnyAttack**, employs a two-stage training paradigm: pre-training and fine-tuning. Figure 2 provides a framework overview of AnyAttack.

In the *pre-training stage*, we train a decoder $F$, to produce adversarial noise $\delta$ on large-scale datasets $\mathcal{D}_p$. Given a batch of images $x$, we extract their embeddings using a frozen image encoder $E$. These normalized embeddings $\mathbf{z}$ are then fed into the decoder $F$, which generates adversarial noise $\delta$ corresponding to the images $x$. To enhance generalization and computational efficiency, we introduce a $K$-augmentation strategy that creates multiple shuffled versions of the original images within each mini-batch. During this process, adversarial noise is added to the shuffled original images (unrelated images) to produce the adversarial images. After passing through $E$, we employ a contrastive loss to maximize the cosine similarity between positive sample pairs (the $i$-th elements of the adversarial and original embeddings) while minimizing the similarity between negative sample pairs (the remaining elements). This approach trains the decoder $F$ to ensure that the perturbed images resemble the original images in the embedding space of encoder $E$, while distinguishing them from the shuffled versions.

In the *fine-tuning stage*, we adapt the pre-trained decoder $F$ to a specific downstream dataset $\mathcal{D}_f$. The frozen encoder $E$ continues to provide embeddings that guide the generation of adversarial noise $\delta$. We use an unrelated random image $x_r$ from an external dataset $\mathcal{D}_e$ as the clean image to synthesize the adversarial image $x_r + \delta$. Unlike the pre-training stage, where only the encoder is utilized as a surrogate model, we introduce auxiliary models during the fine-tuning stage to enhance transferability. We tailor various fine-tuning strategies based on the requirements of each task.

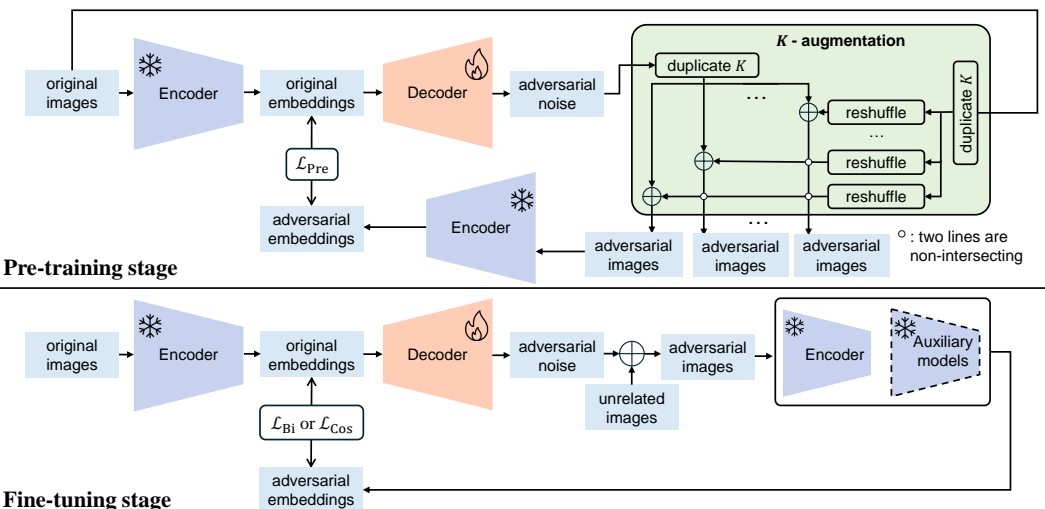

Figure 2: An overview of the proposed AnyAttack.

**Pre-training Stage**  The pre-training phase of AnyAttack aims to train the generator on large-scale datasets, enabling it to handle a diverse array of input images as potential targets. Given a batch of $n$ images $x \in \mathbb{R}^{n \times H \times W \times 3}$ from the large-scale training dataset $\mathcal{D}_p$, we employ the CLIP ViT-B/32 image encoder, which is frozen during training, as the encoder $E$, to obtain the normalized embeddings $E(x) = \mathbf{z} \in \mathbb{R}^{n \times d}$ corresponding to the original images $x$, where $d$ represents the embedding dimension (i.e., 512 for CLIP ViT-B/32). Subsequently, we deploy an initialized decoder $F$, which maps the embeddings $\mathbf{z}$ to adversarial noise $D(\mathbf{z}) = \delta \in \mathbb{R}^{n \times H \times W \times 3}$ corresponding to the original images $x$. We expect the generated noises $\delta$ to serve as adversarial noise representative of the original images $x$. Our goal is for the generated noises $\delta$, when added to random images $x_r$, to be interpreted by the encoder $E$ as the original images $x$, i.e., $E(x_r + \delta) = E(x)$.

However, when the number of random images is smaller than the training dataset, the generated noises $\delta$ may overfit to this limited set, leading to poor generalization for $F$. To address this, we propose the $K$-augmentation strategy, which expands the set of random images to match the size of the training dataset $\mathcal{D}_p$. This strategy increases the number of sample pairs within each batch by a factor of $K$, thereby improving computational efficiency. Specially, $K$-augmentation duplicates both adversarial noises $\delta$ and the original images $x$ $K$ times, forming $K$ mini-batches. For each mini-batch, the order of the adversarial noises remain consistent, while the order of the original images is shuffled within the mini-batch, referred to as shuffled images. These shuffled images are then added to the corresponding adversarial noise, resulting in adversarial images $x'$. Next, the adversarial images are fed into $F$ to produce adversarial embeddings $\mathbf{z}^{(adv)}$, which are then used for subsequent calculations against the original embeddings $\mathbf{z}$. We introduce a contrastive loss that maximizes the cosine similarity between positive sample pairs, defined by the $i$-th elements of adversarial and original embeddings in each mini-batch, while minimizing the similarity between the negative pairs, which consist of all other elements. This setup creates $n$ positive pairs and $n(n-1)$ negative pairs in every mini-batch, with gradients accumulated to update $F$:

$$\mathcal{L}_{\text{Con}} = -\frac{1}{n} \sum_{i=1}^{n} \log \frac{\exp\left(\mathbf{z}_i \cdot \mathbf{z}_i^{(adv)} / \tau(t)\right)}{\sum_{j=1}^{n} \exp\left(\mathbf{z}_i \cdot \mathbf{z}_j^{(adv)} / \tau(t)\right)}, \tag{1}$$

where $\mathbf{z}_i$ and $\mathbf{z}_i^{(adv)}$ are the $\ell_2$-normalized embeddings of the $i$-th sample from original images $x$ and adversarial images $x'$. $\tau(t)$ is the temperature at step $t$, enabling the model to dynamically adjust the hardness of negative samples during training. To facilitate learning and convergence in early training, we set a relatively large initial temperature $\tau_0$ at the beginning of training and gradually decrease it, reaching the final temperature $\tau_{\text{final}}$ after a certain number of steps $T$:

$$\tau(t) = \tau_0 \left(\frac{\tau_{\text{final}}}{\tau_0}\right)^{\frac{t}{T}} = \tau_0 \exp\left(-\lambda t\right). \tag{2}$$

### 3.2.1 FINE-TUNING STAGE

In the fine-tuning stage, we refine the pre-trained decoder $F$ on downstream vision-language datasets using task-specific objective functions, facilitating its adaptation to particular domains and multimodal tasks. The motivation for fine-tuning arises from scenarios where well-defined multimodal tasks and in-domain images are available. Given a batch of $n$ images $x \in \mathbb{R}^{n \times H \times W \times 3}$ from the downstream dataset $\mathcal{D}_f$, the encoder $E$ remains frozen and outputs the embeddings $\mathbf{z}$, which are then fed into the decoder $F$ to generate the noise $\delta$. We randomly select images from an external dataset $\mathcal{D}_e$ as unrelated images $x_r$, which are then added to the generated noise $\delta$ to create adversarial images. To improve transferability, we incorporate auxiliary models alongside the encoder $E$, forming an ensemble surrogate. Drawing on research in ensemble learning for adversarial attacks (Liu et al., 2017), we select auxiliary models based on model diversity, as greater differences between models are known to improve complementarity. This ensures that both adversarial and original images maintain consistency across the embedding spaces of multiple models.

Depending on the downstream tasks, we employ two different fine-tuning objectives. The first strategy is tailored for the image-text retrieval task, which imposes stricter requirements for distinguishing between similar samples. It demands robust retrieval performance in both directions: from $\mathbf{z}^{(adv)}$ to $\mathbf{z}$ and from $\mathbf{z}$ to $\mathbf{z}^{(adv)}$. This motivates the adoption of a bidirectional InfoNCE loss:

$$\mathcal{L}_{\text{Bi}} = \frac{1}{2n} \sum_{i=1}^{n} \left( -\log \frac{\exp(\mathbf{z}_i \cdot \mathbf{z}_i^{(adv)}/\tau)}{\sum_{j=1}^{n} \exp(\mathbf{z}_i \cdot \mathbf{z}_j^{(adv)}/\tau)} - \log \frac{\exp(\mathbf{z}_i^{(adv)} \cdot \mathbf{z}_i/\tau)}{\sum_{j=1}^{n} \exp(\mathbf{z}_i^{(adv)} \cdot \mathbf{z}_j/\tau)} \right). \tag{3}$$

The second strategy is suited for general tasks, such as image captioning, multimodal classification, and other broad vision-language applications. It requires $\mathbf{z}_i^{(adv)}$ to match $\mathbf{z}_i$, so we employ cosine similarity to align $\mathbf{z}_i^{(adv)}$ with $\mathbf{z}_i$, denoting this objective as $\mathcal{L}_{\text{Cos}}$.

## 4 EXPERIMENTS

In this section, we evaluate the performance of our proposed attack across multiple datasets, tasks, and VLMs. We evaluate the effectiveness of targeted adversarial attacks first in image-text retrieval tasks, then multimodal classification tasks, and finally image captioning tasks. Additionally, we analyze the performance of targeted adversarial images on commercial VLMs.

### 4.1 EXPERIMENTAL SETUP

**Baselines** We first employed the state-of-the-art (SOTA) targeted adversarial attack for VLMs, referred to as AttackVLM (Zhao et al., 2024). This method includes two variations: AttackVLM-ii and AttackVLM-it, which are based on different attack objectives. Both methods utilize the CLIP ViT-B/32 image encoder as the surrogate model, consistent with our approach. Additionally, we incorporated two targeted adversarial attacks designed for visual classification models: SU (Wei et al., 2023) and SASD-WS (Wu et al., 2024). Since the original cross-entropy loss used in these methods is not suitable for vision-language tasks, we modified them to employ cosine loss and mean squared error (MSE) loss to match targeted images. These modified methods are denoted as SU-Cos/SASD-WS-Cos and SU-MSE/SASD-WS-MSE, respectively. For the SU attack, the surrogate model is configurable, and we set it to align with our proposed method, namely, CLIP ViT-B/32. For the SASD-WS attack, we utilized the officially released weights, as its surrogate model includes a self-enhancement component. We denote our proposed methods as AnyAttack-Cos, AnyAttack-Bi, AnyAttack-Cos w/ Aux, and AnyAttack-Bi w/ Aux. These represent AnyAttack fine-tuned with $\mathcal{L}_{\text{Cos}}$, fine-tuned with $\mathcal{L}_{\text{Bi}}$, fine-tuned with $\mathcal{L}_{\text{Cos}}$ using auxiliary models, and fine-tuned with $\mathcal{L}_{\text{Bi}}$ using auxiliary models, respectively.

**Datasets, Models, and Tasks** For the downstream datasets, we utilize the MSCOCO, Flickr30K, and SNLI-VE datasets. We employ a variety of target models, including CLIP, BLIP, BLIP2, InstructBLIP, and MiniGPT-4. The downstream tasks we focus on are image-text retrieval, multimodal classification, and image captioning. For each task, we selected the top 1,000 images. Additionally, following the methodology outlined in (Zhao et al., 2024), we used the top 1,000 images from the ImageNet-1K validation set as clean (unrelated) images to generate adversarial examples.

Table 2: The retrieval performances on the MSCOCO dataset under different attacks. TR@1, TR@5, and TR@10 measures text retrieval performance, while IR@1, IR@5, and IR@10 measures image retrieval performance. R@Mean is the average of all retrieval metrics. Our proposed methods are *italicized*, the best results are highlighted in **bold**, and the second-best results are underlined.

| Attack Method | MSCOCO | | | | | | |
|---|---|---|---|---|---|---|---|
| | TR@1 | TR@5 | TR@10 | IR@1 | IR@5 | IR@10 | R@Mean |
| **ViT-B/16** | | | | | | | |
| AttackVLM-ii | 0.4 | 1.0 | 1.4 | 0.24 | 1.08 | 2.16 | 1.05 |
| AttackVLM-it | 0.2 | 1.4 | 1.8 | 0.16 | 1.16 | 2.12 | 1.14 |
| SASD-WS-Cos | 6.0 | 17.0 | 24.8 | 9.08 | 24.39 | 34.55 | 19.30 |
| SASD-WS-MSE | 4.8 | 18.4 | 25.6 | 8.20 | 25.87 | 35.15 | 19.67 |
| SU-Cos | 6.8 | 20.4 | 27.8 | 11.11 | 25.70 | 33.34 | 20.86 |
| SU-MSE | 6.8 | 20.6 | 27.0 | 10.83 | 25.10 | 32.62 | 20.49 |
| *AnyAttack-Cos* | 8.6 | 21.2 | 29.6 | 10.80 | 27.59 | 37.50 | 22.55 |
| *AnyAttack-Bi* | 12.2 | 26.2 | 33.8 | 12.63 | 31.71 | 40.86 | 26.23 |
| *AnyAttack-Cos w/ Aux* | 8.4 | 24.8 | 33.0 | 11.59 | 32.10 | 44.98 | 25.81 |
| *AnyAttack-Bi w/ Aux* | **14.8** | **36.8** | **48.0** | **17.59** | **42.02** | **56.05** | **35.88** |
| **ViT-L/14** | | | | | | | |
| AttackVLM-ii | 0.2 | 1.0 | 1.6 | 0.24 | 0.60 | 1.32 | 0.83 |
| AttackVLM-it | 0.4 | 0.8 | 1.4 | 0.12 | 0.76 | 1.48 | 0.83 |
| SASD-WS-Cos | 3.8 | 11.6 | 18.8 | 7.20 | 18.43 | 26.35 | 14.36 |
| SASD-WS-MSE | 5.4 | 14.6 | 20.6 | 6.00 | 18.23 | 26.47 | 15.22 |
| SU-Cos | 3.0 | 10.4 | 13.2 | 6.19 | 14.99 | 20.07 | 11.31 |
| SU-MSE | 3.4 | 11.2 | 17.4 | 6.63 | 15.27 | 19.79 | 12.28 |
| *AnyAttack-Cos* | 3.8 | 14.0 | 22.8 | 7.36 | 20.71 | 27.55 | 16.04 |
| *AnyAttack-Bi* | 4.8 | 16.0 | 23.6 | 8.20 | 22.31 | 29.11 | 17.34 |
| *AnyAttack-Cos w/ Aux* | 9.4 | 24.6 | 37.0 | 11.51 | 32.62 | 48.18 | 27.22 |
| *AnyAttack-Bi w/ Aux* | **12.0** | **34.0** | **47.4** | **15.67** | **39.34** | **53.54** | **33.66** |
| **ViT-L/14 × 336** | | | | | | | |
| AttackVLM-ii | 0.2 | 0.6 | 1.6 | 0.16 | 1.12 | 2.04 | 0.95 |
| AttackVLM-it | 0.2 | 0.6 | 1.8 | 0.32 | 0.96 | 1.76 | 0.94 |
| SASD-WS-Cos | 2.8 | 10.8 | 16.4 | 6.52 | 18.31 | 26.19 | 13.50 |
| SASD-WS-MSE | 4.4 | 13.6 | 19.2 | 6.72 | 18.23 | 25.71 | 14.64 |
| SU-Cos | 2.4 | 8.0 | 11.2 | 4.88 | 13.79 | 18.39 | 9.78 |
| SU-MSE | 3.6 | 8.2 | 13.2 | 6.40 | 14.51 | 19.19 | 10.85 |
| *AnyAttack-Cos* | 4.6 | 11.0 | 16.6 | 5.96 | 17.67 | 24.23 | 13.34 |
| *AnyAttack-Bi* | 3.6 | 14.4 | 19.0 | 7.64 | 19.79 | 26.83 | 15.21 |
| *AnyAttack-Cos w/ Aux* | 9.0 | 23.2 | 37.2 | 11.68 | 34.03 | 47.62 | 27.12 |
| *AnyAttack-Bi w/ Aux* | **12.0** | **33.2** | **46.8** | **14.79** | **39.22** | **53.06** | **33.18** |

**Metric** In this work, we examine perturbations constrained by the $\ell_\infty$ norm, ensuring that the perturbation $\delta$ satisfies the condition $\|\delta\|_\infty \leq \epsilon$, where $\epsilon$ defines the maximum allowable magnitude of the perturbation. We use the attack success rate (ASR) as the primary evaluation metric to assess the performance of targeted adversarial attacks. The calculation of ASR varies slightly depending on the specific task. For instance, in image-text retrieval tasks, ASR represents the recall rate between adversarial images and their corresponding ground-truth text descriptions. In multimodal classification tasks, ASR refers to the accuracy of correctly classifying pairs of "adversarial image and ground-truth description." Essentially, ASR is calculated by replacing clean images with their adversarial counterparts and then computing the relevant task-specific evaluation metric.

**Implementation Details** We pre-trained the decoder for 520,000 steps on the LAION-400M dataset (Schuhmann et al., 2021), using a batch size of 600 per GPU on three NVIDIA A100 80GB GPUs. The optimizer used was AdamW, with an initial learning rate of $1 \times 10^{-4}$, which was adjusted using cosine annealing. For the downstream datasets, we fine-tuned the decoder for 20 epochs using the same optimizer, initial learning rate, and cosine annealing schedule. We deployed two auxiliary models: the first is a ViT-B/16 trained from scratch on ImageNet-1K, utilizing official weights from torchvision, and the second is the EVA model (Fang et al., 2023; 2024), a ViT-L/14 model that employs masked image modeling and has been fine-tuned on ImageNet-1K. The maximum perturbation $\epsilon$ for $\delta$ was set to $\frac{16}{255}$, and $K$ was set to 5. In the pre-training stage, the initial temperature $\tau_0$ was

set to 1, the final temperature $\tau_{\text{final}}$ was set to 0.07, and the total steps $T$ were set to 10,000. In the fine-tuning stage, the temperature $\tau$ was fixed at 0.07. More details can be found in the Appendix.

## 4.2 EVALUATION ON IMAGE-TEXT RETRIEVAL

In this subsection, we compare the performance of our method against baseline approaches on the image-text retrieval task. Table 2 presents the results on the MSCOCO dataset, while results on the Flickr30K dataset are detailed in the Appendix. The following key observations can be made:

- **Performance of AnyAttack-Bi w/ Auxiliary**: This variant achieves significantly superior performance compared to all baselines, surpassing the best-performing baseline by 15.02%, 18.44%, and 18.54% on ViT-B/16, ViT-B/32, and ViT-L/14, respectively. All AnyAttack methods consistently deliver competitive results, outperforming most baselines. This highlights the effectiveness of our proposed method.

- **Effectiveness of the Auxiliary Module**: The Auxiliary module demonstrates its effectiveness, providing improvements of 6.455%, 13.75%, and 15.875% on ViT-B/16, ViT-B/32, and ViT-L/14, respectively, when comparing AnyAttack w/ Auxiliary to AnyAttack.

- **Advantages of Bidirectional InfoNCE Loss**: The bidirectional InfoNCE loss $\mathcal{L}_{\text{Bi}}$ shows clear advantages for retrieval tasks, with AnyAttack-Bi consistently outperforming AnyAttack-Cos.

## 4.3 EVALUATION ON MULTIMODAL CLASSIFICATION

Here, we compare the performance of our attack with the baselines on the multimodal classification task. Table 3 presents the results on the SNLI-VE dataset. Our method, AnyAttack-Cos w/ Auxiliary, achieves the highest performance, surpassing the strongest baseline, SASD-WS-MSE, by 20.0%. This underscores the effectiveness of our attack in multimodal classification tasks.

Table 3: Attack performance comparison on the SNLI-VE dataset for multimodal classification.

| Attack Method | Accuracy |
|---|---|
| AttackVLM-ii | 6.5 |
| AttackVLM-it | 6.3 |
| SASD-WS-Cos | 24.3 |
| SASD-WS-MSE | 24.8 |
| SU-Cos | 13.7 |
| SU-MSE | 13.6 |
| *AnyAttack-Cos* | 17.5 |
| *AnyAttack-Cos w/ Aux* | **44.8** |

## 4.4 EVALUATION ON IMAGE CAPTIONING

Here, we evaluate the performance of our attack on the image captioning task using the MSCOCO dataset. The VLMs take adversarial images as input and generate text descriptions, which are then assessed against the ground-truth captions using standard metrics. Table 4 presents the results across four VLMs: InstructBLIP, BLIP2, BLIP, and MiniGPT-4. Our attack AnyAttack-Cos w/ Auxiliary consistently demonstrates superior performance across all evaluation metrics, outperforming the baseline attacks on each VLM.

## 4.5 TRANSFER TO COMMERCIAL VLMS

Here, we transfer the targeted adversarial images generated by the pre-trained decoder (referred to as AnyAttack-Pre) to three commercial VLMs, including Claude's Sonnet, Microsoft's Copilot, and Google's Gemini. We utilized the publicly available web interfaces of these models. Figure 3 illustrates the example responses of the three commercial models, with more examples are provided in the Appendix. No prior conversation context or constraints were applied; the only prompt used was "Describe this image." The portions of the VLM responses highlighted in red correspond to the target images, showcasing the effectiveness of our attack.

## 4.6 FURTHER ANALYSIS

**Ablation Study** We perform an ablation study on the MSCOCO dataset for the image-text retrieval task to evaluate the impact of three key components in our approach: 1) **Training approach**: Pre-trained, fine-tuned, or trained from scratch. 2) **Auxiliary models**: With or without auxiliary model integration. 3) **Fine-tuning objective**: Cosine similarity loss ($\mathcal{L}_{\text{Cos}}$) vs. bidirectional contrastive loss ($\mathcal{L}_{\text{Bi}}$).

Table 4: Attack performance comparison on the MSCOCO dataset for image captioning task.

| Model | Attack Method | SPICE | BLEU-1 | BLEU-4 | METEOR | ROUGE-L | CIDEr |
|---|---|---|---|---|---|---|---|
| InstructBLIP | AttackVLM-ii | 1.4 | 38.9 | 5.4 | 8.7 | 28.6 | 3.4 |
| | AttackVLM-it | 1.3 | 39.1 | 5.4 | 8.7 | 28.8 | 3.3 |
| | SASD-WS-Cos | 3.4 | 43.9 | 7.2 | 10.5 | 32.2 | 10.9 |
| | SASD-WS-MSE | 3.2 | 44.6 | 7.0 | 10.8 | 32.4 | 11.8 |
| | SU-Cos | 1.9 | 40.7 | 6.0 | 9.3 | 29.9 | 5.3 |
| | SU-MSE | 1.9 | 40.9 | 6.5 | 9.5 | 29.9 | 6.0 |
| | *AnyAttack-Cos* | 2.3 | 41.5 | 5.9 | 9.5 | 30.2 | 7.0 |
| | *AnyAttack-Cos w/ Aux* | **4.7** | **46.5** | **7.5** | **12.2** | **33.6** | **20.3** |
| BLIP2 | AttackVLM-ii | 1.2 | 39.6 | 5.3 | 8.7 | 29.0 | 3.6 |
| | AttackVLM-it | 1.2 | 39.6 | 5.4 | 8.7 | 29.3 | 3.5 |
| | SASD-WS-Cos | 2.6 | 43.0 | 6.3 | 10.2 | 32.0 | 9.3 |
| | SASD-WS-MSE | 2.8 | 42.8 | 6.5 | 10.2 | 31.7 | 9.5 |
| | SU-Cos | 1.6 | 40.9 | 5.6 | 9.2 | 30.1 | 4.7 |
| | SU-MSE | 1.6 | 40.8 | 5.9 | 9.2 | 30.1 | 5.0 |
| | *AnyAttack-Cos* | 1.8 | 41.3 | 5.2 | 9.6 | 30.9 | 5.6 |
| | *AnyAttack-Cos w/ Aux* | **3.3** | **44.2** | **6.0** | **11.0** | **32.4** | **13.3** |
| BLIP | AttackVLM-ii | 1.3 | 39.8 | 5.0 | 8.8 | 29.9 | 3.4 |
| | AttackVLM-it | 1.2 | 39.7 | 4.8 | 8.7 | 29.7 | 3.2 |
| | SASD-WS-Cos | 3.3 | 43.8 | 6.9 | 10.7 | 32.3 | 11.9 |
| | SASD-WS-MSE | **3.4** | 43.8 | 6.9 | 10.8 | 32.3 | 12.4 |
| | SU-Cos | 2.6 | 43.0 | 6.5 | 10.1 | 31.8 | 8.7 |
| | SU-MSE | 2.6 | 42.4 | 6.4 | 9.9 | 31.6 | 8.4 |
| | *AnyAttack-Cos* | 2.2 | 41.6 | 6.0 | 9.5 | 31.1 | 6.1 |
| | *AnyAttack-Cos w/ Aux* | 3.4 | **44.4** | **7.1** | **11.1** | **32.8** | **13.0** |
| Mini-GPT4 | AttackVLM-ii | 1.6 | 29.5 | 2.3 | 9.3 | 24.3 | 1.6 |
| | AttackVLM-it | 1.5 | 29.2 | 2.3 | 9.4 | 24.5 | 1.5 |
| | SASD-WS-Cos | 2.8 | 30.5 | 2.4 | 10.9 | 25.2 | 2.6 |
| | SASD-WS-MSE | 3.1 | 30.5 | 2.9 | 10.9 | 25.7 | 2.8 |
| | SU-Cos | 2.0 | 29.5 | 2.9 | 9.9 | 24.8 | 1.9 |
| | SU-MSE | 2.2 | 30.3 | 2.9 | 9.9 | 25.1 | 2.2 |
| | *AnyAttack-Cos* | 2.5 | 30.5 | 2.4 | 10.3 | 25.2 | 1.9 |
| | *AnyAttack-Cos w/ Aux* | **4.6** | **32.5** | **4.0** | **12.4** | **27.3** | **5.3** |

The results, summarized in Figure 4, reveal the following: 1) **Training approach**: Fine-tuning a pre-trained model achieves the highest performance, while training from scratch yields significantly worse results, indicating that pre-training is critical for task adaptation. 2) **Auxiliary models**: The inclusion of auxiliary models consistently improves performance, highlighting their role in enhancing transferability. 3) **Fine-tuning objective**: The bidirectional contrastive loss ($\mathcal{L}_{\text{Bi}}$) consistently outperforms the cosine similarity loss ($\mathcal{L}_{\text{Cos}}$), demonstrating its effectiveness in improving the alignment of image and text embeddings.

**Efficiency Analysis** In this subsection, we compare the efficiency of our method with SU, SASD, and AttackVLM. Figure 5 presents the results for generating 1,000 adversarial images on a single NVIDIA A100 80GB GPU with a batch size of 250, showing both memory usage and time consumption. The results

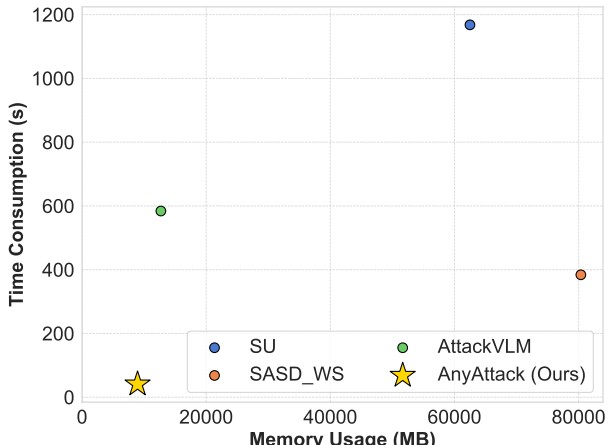

Figure 5: Efficiency Analysis: Memory Usage vs Time Consumption.

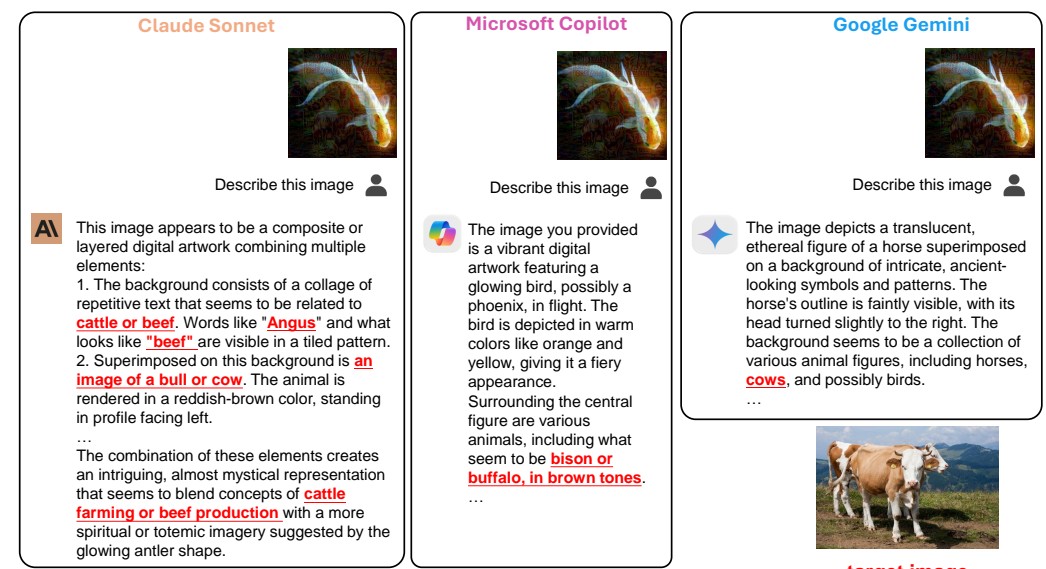

Figure 3: Example responses from 3 commercial VLMs to targeted attacks generated by our method.

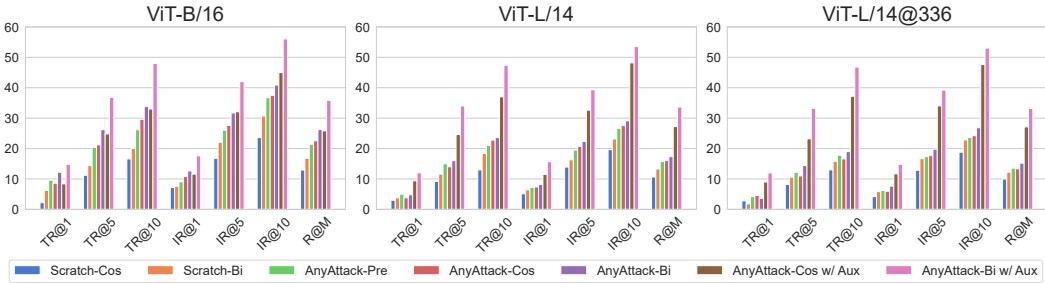

Figure 4: Performance comparison between different configurations of AnyAttack for the image-text retrieval task on MSCOCO. The plot shows the comparative performance of models initialized from scratch, pre-trained, and fine-tuned, alongside the impact of auxiliary models and different fine-tuning objectives on retrieval tasks.

demonstrate that our approach significantly outperforms the baselines in both computational speed and memory efficiency.

## 5 CONCLUSION

In this paper, we introduced **AnyAttack**, a novel self-supervised framework for generating targeted adversarial attacks on VLMs. Our approach overcomes the scalability limitations of previous methods by enabling the use of **any** image to serve as a target for **attack** target without label supervision. Through extensive experiments, we demonstrated the effectiveness of AnyAttack across multiple VLMs and vision-language tasks, revealing significant vulnerabilities in state-of-the-art models. Notably, our method showed considerable transferability, even to commercial VLMs, highlighting the broad implications of our findings.

These results underscore the urgent need for robust defense mechanisms in VLM systems. As VLMs become increasingly prevalent in real-world applications, our work opens new avenues for research in VLM security, particularly considering that this is the first time pre-training has been conducted on a large-scale dataset like LAION-400M. This emphasizes the critical importance of addressing these challenges. Future work should focus on developing resilient VLMs and exploring potential mitigation strategies against such targeted attacks.

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

## A  APPENDIX

In this appendix, we describe more implementation details and additional experiment results.

### A.1  IMPLEMENTATION DETAILS

Table 5: Configuration details of VLMs.

| Model | Configuration |
|---|---|
| BLIP2 | `caption_coco_opt2.7b` |
| InstructBLIP | `vicuna-7b-v1.1` |
| Mini-GPT4 | `minigpt4_llama2_7b` |

In this section, we provide additional details regarding the experimental setup. For AttackVLM, we utilized the official code[2], while for SU and SASD-WS, we employed the TransferAttack tool[3]. For

---

[2] `https://github.com/yunqing-me/AttackVLM`.

[3] `https://github.com/Trustworthy-AI-Group/TransferAttack`.

Table 6: Comparison of retrieval performance on the Flickr30k dataset. TR@1, TR@5, and TR@10 represent text retrieval, while IR@1, IR@5, and IR@10 represent image retrieval. R@Mean is the average of all retrieval metrics. Our proposed methods are *italicized*, the best results are highlighted in **bold**, and the second-best results are underlined.

| | Method | Flickr30k | | | | | | |
|---|---|---|---|---|---|---|---|---|
| | | TR@1 | TR@5 | TR@10 | IR@1 | IR@5 | IR@10 | R@Mean |
| ViT-B/16 | AttackVLM-ii | 0.2 | 1.8 | 2.4 | 1.00 | 2.00 | 2.80 | 1.70 |
| | AttackVLM-it | 0.4 | 1.6 | 2.6 | 0.20 | 1.60 | 3.00 | 1.57 |
| | SASD-WS-Cos | 4.4 | 12.6 | 19.8 | 7.00 | 17.20 | 25.20 | 14.37 |
| | SASD-WS-MSE | 4.2 | 12.2 | 18.0 | 5.20 | 16.00 | 23.00 | 13.10 |
| | SU-Cos | 5.0 | 17.2 | 25.2 | 8.40 | 20.20 | 27.00 | 17.17 |
| | SU-MSE | 5.8 | 15.8 | 24.2 | 9.40 | 21.60 | 30.20 | 17.83 |
| | *AnyAttack-Cos* | 7.0 | 18.8 | 24.8 | 11.40 | 25.40 | 33.00 | 20.07 |
| | *AnyAttack-Bi* | 8.8 | 22.4 | 30.0 | 11.80 | 27.20 | 37.20 | 22.90 |
| | *AnyAttack-Cos w/ Aux* | 8.8 | 24.4 | 35.6 | 15.40 | 31.60 | 40.40 | 26.03 |
| | *AnyAttack-Bi w/ Aux* | **14.6** | **32.2** | **44.2** | **19.00** | **36.60** | **45.00** | **31.93** |
| ViT-L/14 | AttackVLM-ii | 0.4 | 1.0 | 2.6 | 0.60 | 1.20 | 1.80 | 1.27 |
| | AttackVLM-it | 0.6 | 0.8 | 2.8 | 0.60 | 1.20 | 2.40 | 1.40 |
| | SASD-WS-Cos | 1.8 | 9.2 | 13.0 | 3.40 | 11.40 | 17.40 | 9.37 |
| | SASD-WS-MSE | 3.0 | 7.8 | 15.8 | 4.00 | 10.60 | 17.60 | 9.80 |
| | SU-Cos | 2.0 | 8.4 | 14.2 | 5.20 | 13.80 | 19.60 | 10.53 |
| | SU-MSE | 2.0 | 7.6 | 12.8 | 4.00 | 12.80 | 15.60 | 9.13 |
| | *AnyAttack-Cos* | 4.6 | 11.8 | 18.2 | 10.60 | 20.20 | 25.60 | 15.17 |
| | *AnyAttack-Bi* | 5.4 | 15.8 | 21.8 | 10.20 | 22.40 | 29.40 | 17.50 |
| | *AnyAttack-Cos w/ Aux* | 7.8 | 22.8 | 33.0 | 13.80 | 29.20 | 38.40 | 24.16 |
| | *AnyAttack-Bi w/ Aux* | **12.4** | **30.6** | **41.0** | **15.20** | **34.60** | **44.80** | **29.77** |
| ViT-L/14 × 336 | AttackVLM-ii | 0.4 | 0.8 | 2.4 | 0.40 | 1.60 | 2.00 | 1.27 |
| | AttackVLM-it | 0.6 | 1.0 | 2.4 | 0.00 | 1.60 | 2.20 | 1.30 |
| | SASD-WS-Cos | 3.0 | 8.4 | 13.8 | 4.40 | 14.00 | 19.20 | 10.47 |
| | SASD-WS-MSE | 3.0 | 9.0 | 14.4 | 3.60 | 11.60 | 19.00 | 10.10 |
| | SU-Cos | 2.6 | 7.4 | 10.6 | 5.20 | 11.60 | 15.80 | 8.87 |
| | SU-MSE | 3.0 | 8.0 | 12.2 | 5.60 | 11.80 | 17.40 | 9.67 |
| | *AnyAttack-Cos* | 4.6 | 10.0 | 14.6 | 8.00 | 17.40 | 22.60 | 12.87 |
| | *AnyAttack-Bi* | 4.2 | 10.8 | 17.4 | 8.20 | 19.00 | 26.20 | 14.30 |
| | *AnyAttack-Cos w/ Aux* | 8.0 | 20.2 | 30.4 | 13.80 | 30.40 | 38.40 | 21.87 |
| | *AnyAttack-Bi w/ Aux* | **9.8** | **26.0** | **34.2** | **16.20** | **34.60** | **45.00** | **27.63** |

VLMs integrated with LLMs, we used the LAVIS library[4] and the MiniGPT-4 repository[5]. More details are provided in Table 5. Regarding MiniGPT-4, it tends to generate detailed responses, even when the prompt "Describe this image in one short sentence only". Occasionally, it outputs multiple sentences, which affects its scoring in the image captioning task.

## A.2 ADDITIONAL EXPERIMENT RESULTS

**Additional Results on Image-text Retrieval**   We also report the retrieval performance of our method and the baseline methods on the Flickr30k dataset, with results shown in Table 6. The conclusions are consistent with those obtained from the MSCOCO dataset.

**Additional Results on Image Captioning**   We present additional examples of the Image Captioning task in Figures 6, 7, and 8. The visualized results further demonstrate the effectiveness of our approach. Interestingly, MiniGPT-4 models (including MiniGPT-v2) tend to generate longer responses. While this behavior slightly affects their quantitative performance on image captioning metrics (as shown in Table 4), the qualitative results suggest that their output remains quite effective.

---

[4]`https://github.com/salesforce/LAVIS.`
[5]`https://github.com/Vision-CAIR/MiniGPT-4.`

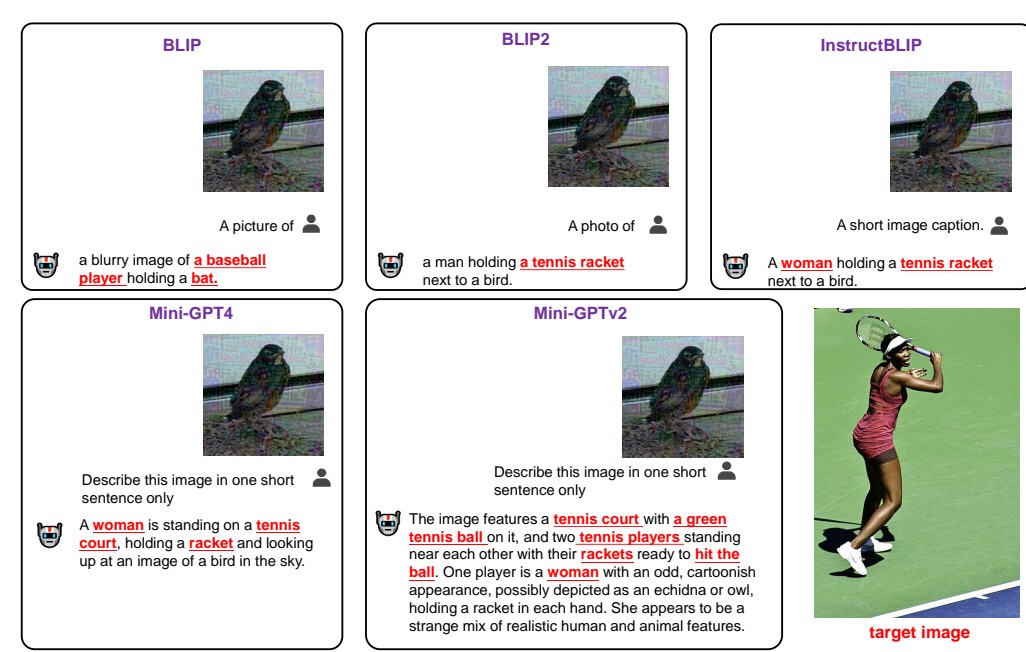

Figure 6: Visual examples of the image captioning task (Example 1).

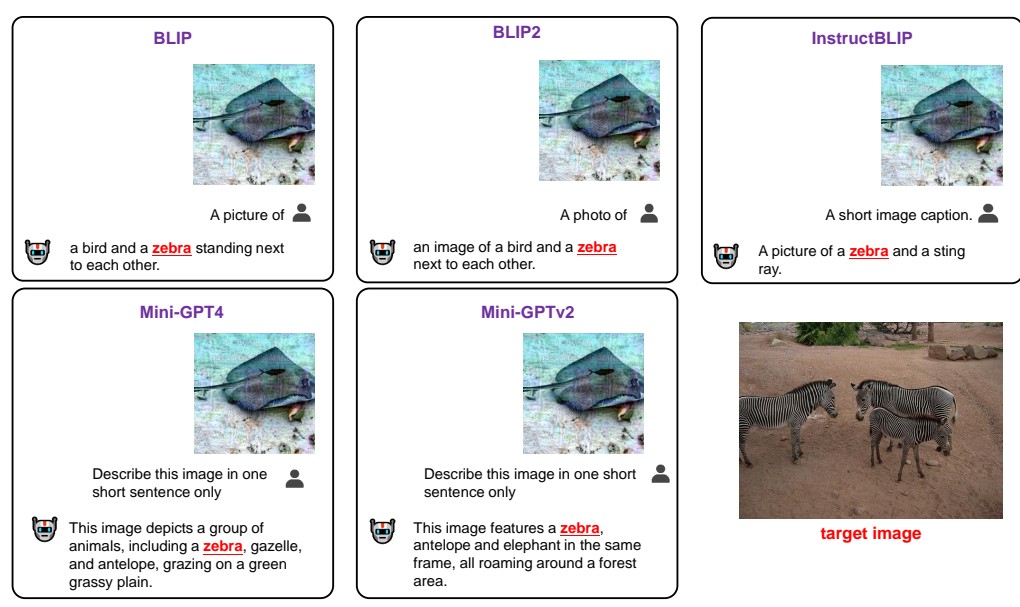

Figure 7: Visual examples of the image captioning task (Example 2).

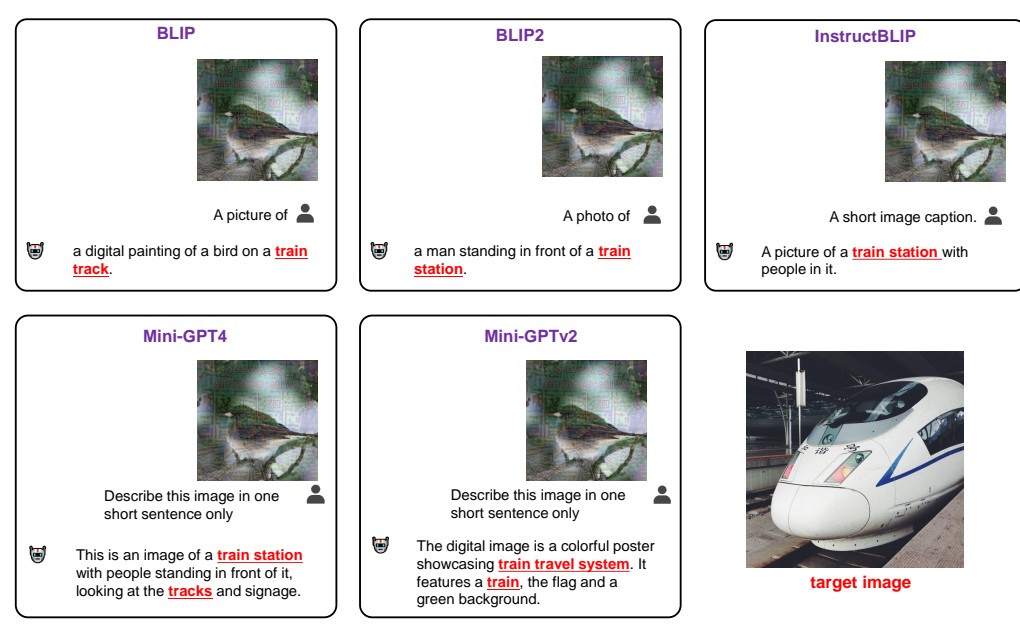

Figure 8: Visual examples of the image captioning task (Example 3).

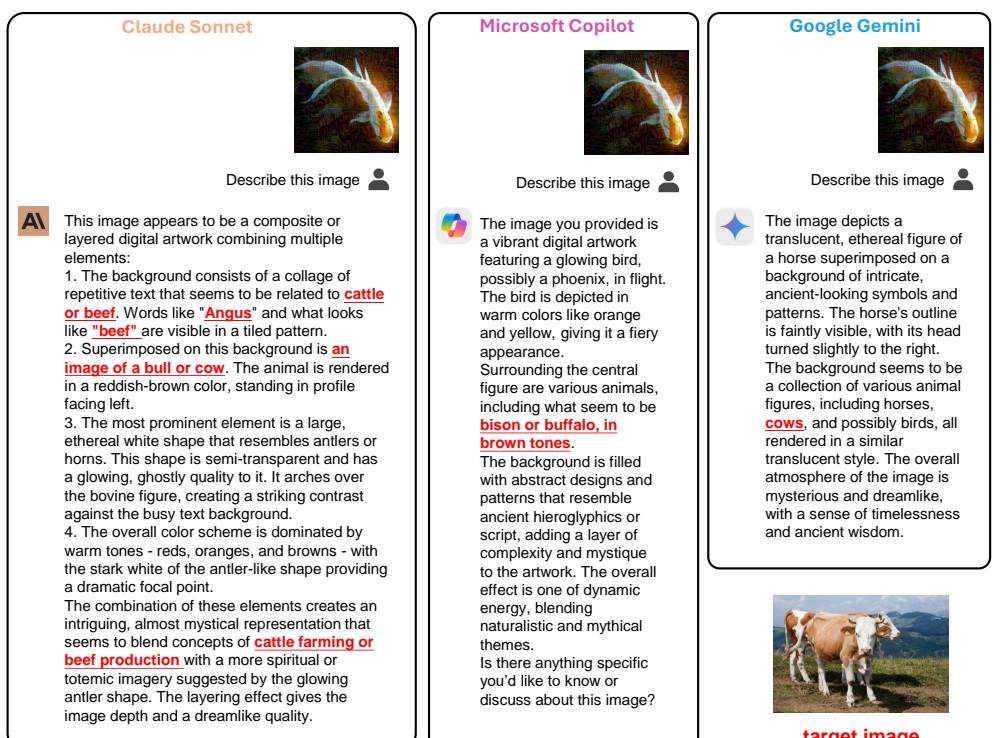

Figure 9: Responses from three commercial VLMs to our targeted adversarial image (Example 1).

**Additional Results on Commercial VLMs**  To further demonstrate the effectiveness of our method, we present additional examples in Figures 9, 10, and 11, where we transfer our attack to commercial VLMs, including Claude's Sonnet, Microsoft's Copilot, and Google's Gemini. The adversarial images generated by our pre-trained decoder were uploaded to these models using their

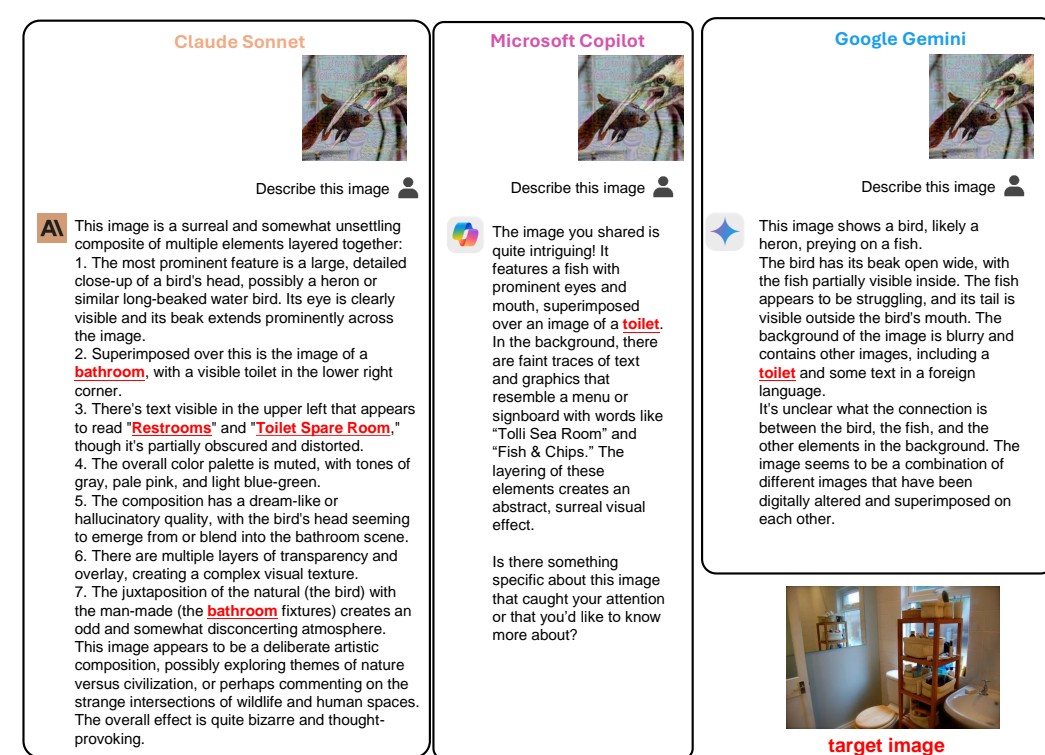

Figure 10: Responses from three commercial VLMs to our targeted adversarial image (Example 2).

publicly available web interfaces. No prior context or constraints were provided, and the only prompt used was "Describe this image".

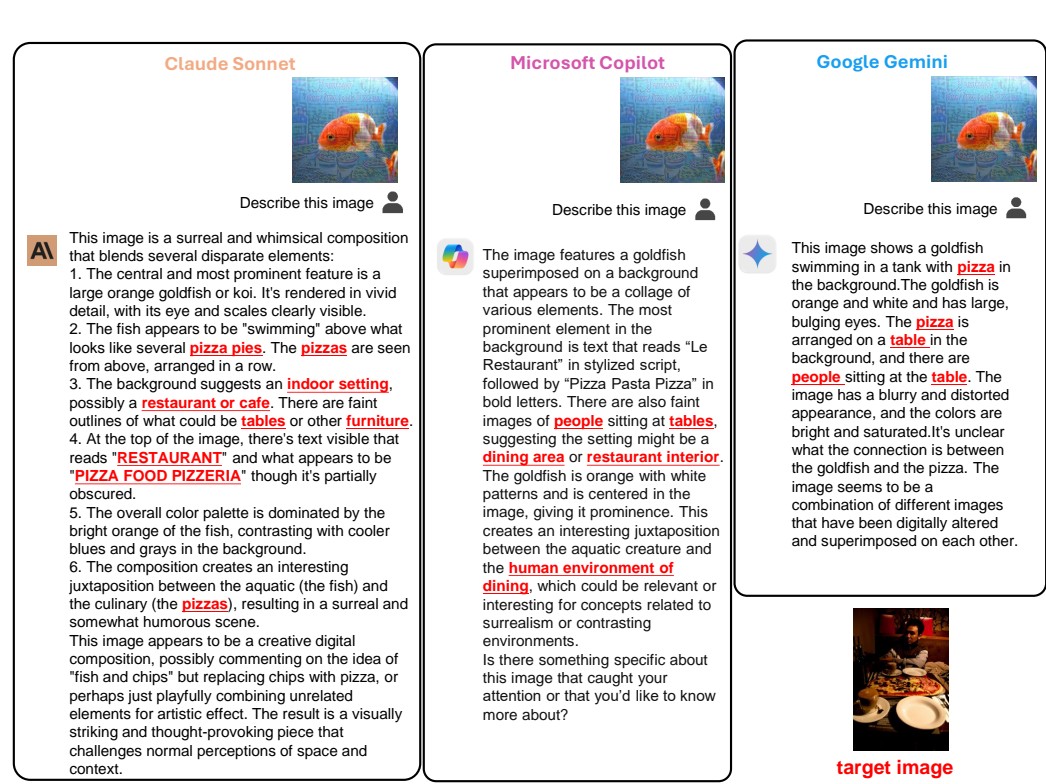

Figure 11: Responses from three commercial VLMs to our targeted adversarial image (Example 3).

