# OpenReview forum: "AnyAttack: Self-supervised Generation of Targeted Adversarial Attacks for Vision-Language Models"
_ICLR.cc/2025/Conference — ICLR 2025 Conference Withdrawn Submission_

### Official Review · Reviewer_hQn5 · 2024-10-23

**Soundness:** 2
**Presentation:** 2
**Contribution:** 3
**Rating:** 5
**Confidence:** 3

**Summary:**

The paper propose a self-supervised framework that generates targeted adversarial images for VLMs without label supervision  allowing any image to serve as a target for the attac and  demonstrate the effectiveness of our AnyAttack on five mainstream open-source VLMs
across three multimodal tasks.

**Strengths:**

1. The Proposed method demonstrate strong performance compare to other baseline methods
2. The introduction of self-supervision can effectively overcome the limitation of existing methods that require label supervision.
3. The proposed method is efficient in both memory and time consumption, making it suitable for practical applications.

**Weaknesses:**

Despite the proposed method's effectiveness, there are several areas where this paper could see improvements, including the presentation clarity,experiment setting and theoretical analysis:
1. Figure clarity:In Figure 2 pre-training stage, L_pre is used, however this loss is not explained in figure caption or main text. I guess it refer to contrastive loss. Please explain here for clarity.

2. Experiment setting may not fair: The comparsion between SASD-WS not fair. Anyattack are trained on LAION-400M  while SASD-WS is trained on the significantly smaller ImageNet dataset. The difference in the scale and diversity of the training datasets likely contributes to the performance gap. The author should conduct necessary experiment to make a fair comparsion.

3. The improved performance seems mostly come from the ensemble-based approach [1]. This reliance on an ensemble approach may reduce the effectiveness of the proposed AnyAttack, as the Bi-loss and cos-loss without the auxillary model does not consistently outperform existing methods. Furthermore, since the incorporation of auxiliary models contributed significantly to the overall performance, the authors should provide a more detailed explanation of this in Section 3.2.1. and disclose the hyperparameters used for the ensemble to improve clarity and soundess.

4. lack of theoretical analysis: the paper does not provide a formal theoretical foundation analysis on proposed method.






[1] Liu, Y., Chen, X., Liu, C., & Song, D. (2017). Delving into transferable adversarial examples and black-box attacks. In 5th International Conference on Learning Representations (ICLR).

**Questions:**

1. The metric shown in table 3 used accuracy is weirdo. For attack method, lower accuracy should represent better attack performance. Please clarify here.

Others see the weakness. I appreciate the authors' efforts in introducing a self-supervised adversarial attack framework, however the experiment section and method presentation make me question the soundness of the proposed method.I will adjust my score according to author's response and other reviewer's opinion.

---

### Official Review · Reviewer_euFD · 2024-11-01

**Soundness:** 3
**Presentation:** 2
**Contribution:** 2
**Rating:** 3
**Confidence:** 5

**Summary:**

This paper proposes AnyAttack, a method for generating targeted adversarial images for Vision-Language Models (VLMs) without label supervision. Leveraging K-augmentation during training on a large-scale unlabeled dataset and fine-tuning with task-related loss on downstream tasks, the decoder effectively maps embeddings to adversarial noise, resulting in a higher attack success rate.

**Strengths:**

1. The pipeline is intuitive and easy to understand.
2. The authors conduct extensive experiments across various VLMs and downstream tasks to illustrate the improvement.

**Weaknesses:**

1. The paper contains several writing errors. For instance, in Figure 1 (b), “Our AnyAttack” is incorrectly written as “Our AnyAttak”. Additionally, in line 270, “FINE-TUNING STAGE” should not be formulated as a sub-subsection, to maintain consistency with the “Pre-training Stage” in line 236. Furthermore, the l∞ norm constraint should be included under “Implementation Details”, rather than “Metric”.
2. The improvement primarily stems from the simple ensembling of encoders, which lacks technique contribution. Additionally, the variant AnyAttack-Cos appears to underperform compared to the baseline method SASD-WS-Cos in both multi-modal classification and image captioning tasks, failing to demonstrate the superiority of the decoder-based approach over previous methods based on adversarial noise optimization.
3. In section 4.5, the effect of transferability to commercial VLMs is presented only through visualizations, without any quantitative metrics. This lack of quantitative indicators reduces the persuasiveness of the results.

**Questions:**

1. In the pre-training stage, what advantage does K-argumentation offer over randomly sampling K batches from the training dataset? The latter approach seems to provide greater diversity and generalization.
2. Provide quantitative results for transferability to commercial VLMs. For instance, query the VLMs using the adversarial image with a question like, “Does the image depict [target sentence]?” and then calculate the Attack Success Rate based on the responses.
3. In Figure 5, does the reported time consumption and memory usage encompass both the pre-training and fine-tuning stages or do they represent only the inference stage, where adversarial images are generated using the fully trained model?

---

### Official Review · Reviewer_ZtiB · 2024-11-02

**Soundness:** 2
**Presentation:** 3
**Contribution:** 2
**Rating:** 5
**Confidence:** 4

**Summary:**

This paper proposes AnyAttack, a self-supervised framework for generating targeted adversarial attacks on Vision-Language Models without requiring target labels. The method introduces a contrastive loss that enables training a generator on large-scale unlabeled datasets to generate targeted adversarial noise.

**Strengths:**

AnyAttack enables the generation of targeted adversarial attacks without the need for target labels, allowing any image to be used as a target for the attack.

**Weaknesses:**

1. The technical contribution primarily combines existing techniques - contrastive learning and adversarial attacks - without evident innovations. The main novelty appears to be applying these known methods at scale to VLMs. The authors don't justify why their approach works better than existing methods beyond empirical results.

2. For the evaluation of commercial models (Gemini, Sonnet, and Copilot), this paper fails to provide specific success rates, number of attack attempts, or detailed attack configurations for these systems. Moreover, the authors overlooked testing on several leading commercial VLMs like GPT-4V, GPT-4o, and Claude 3 Sonnet, which are widely considered more capable than the tested models.

3. The paper lacks some ablation studies exploring the impact, e.g., the choice of the contrastive loss function, architecture decisions, and the effect of the K-augmentation strategy.

**Questions:**

See the above.

---

### Official Review · Reviewer_d6jY · 2024-11-02

**Soundness:** 2
**Presentation:** 3
**Contribution:** 2
**Rating:** 5
**Confidence:** 4

**Summary:**

The paper introduces a new self-supervised method named AnyAttack, aimed at generating adversarial attacks against visual language models (VLMs). This method bypasses traditional label-based supervision by leveraging contrastive loss to train on large-scale, unlabeled datasets, thus enabling effective transfer across various VLMs. Furthermore, the paper demonstrates the method's effectiveness and robustness through experiments on multiple tasks and models.

**Strengths:**

- The proposed AnyAttack method does not require label supervision, allowing it to utilize any image as an attack target. This significantly enhances the flexibility and applicability of the attacks.
- The paper provides a thorough evaluation of AnyAttack, validating the method’s effectiveness.

**Weaknesses:**

- The overall design of the approach is relatively simple, which raises concerns about its level of innovation.
- Additionally, a fundamental premise of machine learning or deep learning is that the data or information adheres to a fixed distribution, though it could also be a complex combination of multiple distributions. However, I do not believe adversarial noise follows any specific distribution, which makes it doubtful whether a neural network can effectively learn adversarial noise. I would like the authors to clarify the rationale here.
- The fine-tuning process requires an external dataset, which may introduce information leakage, making the test results potentially unfair. Moreover, using an external dataset similar to the test data typically yields better results.
- In the context of adversarial attacks, "transferability" has a specific meaning: adversarial samples generated on a surrogate model should be able to attack a target model directly. It is unclear if the paper’s use of "transferability" aligns with this definition, and there is insufficient experimental evidence supporting this claim.

**Questions:**

See weaknesses.

---

### Note · Authors · 2024-11-14

I have read and agree with the venue's withdrawal policy on behalf of myself and my co-authors.